# Boreal conifers maintain carbon uptake with warming despite failure to track optimal temperatures

**Mirindi Eric Dusenge** [1,2,3] ✉, **Jeffrey M. Warren** [4], **Peter B. Reich** [5,6,7], **Eric J. Ward** [8], **Bridget K. Murphy** [3,9,10], **Artur Stefanski** [5], **Raimundo Bermudez** [5], **Marisol Cruz** [11], **David A. McLennan** [4], **Anthony W. King** [4], **Rebecca A. Montgomery** [5], **Paul J. Hanson** [4] & **Danielle A. Way** [3,12,13,14] ✉

Warming shifts the thermal optimum of net photosynthesis ($T_{optA}$) to higher temperatures. However, our knowledge of this shift is mainly derived from seedlings grown in greenhouses under ambient atmospheric carbon dioxide ($CO_2$) conditions. It is unclear whether shifts in $T_{optA}$ of field-grown trees will keep pace with the temperatures predicted for the 21st century under elevated atmospheric $CO_2$ concentrations. Here, using a whole-ecosystem warming controlled experiment under either ambient or elevated $CO_2$ levels, we show that $T_{optA}$ of mature boreal conifers increased with warming. However, shifts in $T_{optA}$ did not keep pace with warming as $T_{optA}$ only increased by 0.26–0.35 °C per 1 °C of warming. Net photosynthetic rates estimated at the mean growth temperature increased with warming in elevated $CO_2$ spruce, while remaining constant in ambient $CO_2$ spruce and in both ambient $CO_2$ and elevated $CO_2$ tamarack with warming. Although shifts in $T_{optA}$ of these two species are insufficient to keep pace with warming, these boreal conifers can thermally acclimate photosynthesis to maintain carbon uptake in future air temperatures.

Photosynthesis is the largest annual carbon flux between the atmosphere and the biosphere[1], taking up ~123 gigatons of carbon per year from the atmosphere[2]. Terrestrial photosynthesis is ~11 times higher than annual anthropogenic $CO_2$ emissions[1], offsetting a significant fraction of anthropogenic $CO_2$ emissions[3,4]. Thus, relatively small changes in terrestrial photosynthesis due to global change drivers, such as warming and drought, could increase the rate of atmospheric $CO_2$ accumulation and associated climate warming predicted by Terrestrial Biosphere Models (TBMs)[5] that are a key component of global climate models.

[1]Department of Biology, Mount Allison University, Sackville, NB E4L 1E4, Canada. [2]Western Centre for Climate Change, Sustainable Livelihoods and Health, Department of Geography and Environment, The University of Western Ontario, London, ON N6G 2V4, Canada. [3]Department of Biology, The University of Western Ontario, London, ON N6A 3K7, Canada. [4]Climate Change Science Institute and Environmental Sciences Division, Oak Ridge National Laboratory, Oak Ridge, TN 37830, USA. [5]Department of Forest Resources, University of Minnesota, Saint Paul, MN 55108, USA. [6]Hawkesbury Institute for the Environment, University of Western Sydney, Penrith, NSW 2753, Australia. [7]Institute for Global Change Biology, and School for the Environment and Sustainability, University of Michigan, Ann Arbor, MI 48109, USA. [8]US Geological Survey, Wetland and Aquatic Research Center, Lafayette, LA, USA. [9]Department of Biology, University of Toronto Mississauga, Mississauga, ON L5L 1C6, Canada. [10]Graduate Program in Cell and Systems Biology, University of Toronto, Toronto, ON M5S 3B2, Canada. [11]Departamento de Ciencias Biologicas, Universidad de Los Andes, Bogota, Colombia. [12]Division of Plant Sciences, Research School of Biology, The Australian National University, Canberra, ACT 2601, Australia. [13]Nicholas School of the Environment, Duke University, Durham, NC 27708, USA. [14]Environmental and Climate Sciences Department, Brookhaven National Laboratory, Upton, NY 11973, USA. ✉e-mail: mdusenge@mta.ca; danielle.way@anu.edu.au

To improve predictions of $CO_2$ exchange between terrestrial vegetation and the atmosphere in the warmer, elevated $CO_2$ climates of the future, it is critical to account for acclimation of photosynthesis to both warming and elevated $CO_2$ within TBMs[6,7]. The photosynthetic temperature sensitivity functions currently employed within TBMs were developed using data largely derived from young trees grown in greenhouse warming experiments under ambient atmospheric $CO_2$ conditions[6,8]. Thus, it is unclear whether these thermal responses accurately represent mature trees growing in natural conditions in the field and whether they hold under elevated atmospheric $CO_2$ conditions.

Photosynthesis is regulated by several types of processes (biochemical, biomechanical and diffusional) which are all temperature dependent[9–11]. In the short-term (minutes to hours), photosynthesis responds non-linearly to temperature, increasing up to a thermal optimum ($T_{optA}$) and decreasing at supra-optimal temperatures. The decrease of photosynthesis at supra-optimal temperatures is caused by various processes including increased membrane fluidity[12,13], impaired redox reactions between protein complexes and electron carriers[14], reduced intracellular $CO_2$ availability due to stomatal closure[15], deactivation of the key photosynthetic enzyme Rubisco (ribulose-1,5-biphosphate carboxylase/oxygenase)[16], and the release of previously-fixed $CO_2$ through high respiration and photorespiration rates[5,9–11]. When exposed to long-term warming (days to years), plants generally acclimate photosynthesis by increasing the $T_{optA}$[8,11,17–23], thereby increasing net carbon uptake at the new warmer temperature. This acclimation to high temperatures can involve decreased thylakoid membrane fluidity[24], expression of a more heat-stable Rubisco[25] and Rubisco activase[11], expression of heat shock proteins[11], and decreases in respiration[26–28]. However plants differ greatly in their ability to thermally acclimate $T_{optA}$, with reported values in the literature ranging from increases in the $T_{optA}$ of 0.16–0.78 °C per 1 °C of warming[8,11,19,22,29–31]. Among the conifers that dominate the boreal forest, some species have shown the ability to acclimate $T_{optA}$[30,32,33] to warming, while others have not[34]. Whether such stark differences in acclimation capacity are truly representative (i.e., do some species acclimate while others do not) or result from modest sampling intensity is as of yet unclear. Moreover, these studies on boreal conifers have been conducted on seedlings in growth chambers and greenhouses, and it is unclear whether these photosynthetic acclimation responses translate to mature trees growing in the variable air temperatures found in the forest. Furthermore, these studies rarely investigate whether increases in $T_{optA}$ match increases in growth temperature. In a three-year field warming study on broad-leaved boreal and temperate seedlings, shifts in $T_{optA}$ occurred but were much smaller than increases in growth temperatures[19]. However, no study to date has explored whether mature field-grown conifers, the trees that represent the majority of the boreal forest, can adjust $T_{optA}$ to compensate for the increasing air temperatures expected over the next few decades.

Photosynthesis and $T_{optA}$ are also affected by elevated $CO_2$. Elevated $CO_2$ concentrations stimulate photosynthesis because $CO_2$ is the substrate for Rubisco[35–37], the carboxylating enzyme in $C_3$ photosynthesis. In the long term, this initial stimulation of photosynthesis often (but not always[38]) diminishes[39] due to acclimation of the photosynthetic biochemistry to elevated $CO_2$ concentrations and plant sink limitations[36,40,41]. In some instances, the initial stimulation of photosynthesis by high $CO_2$ completely disappears, mainly due to nitrogen limitation[42]. By increasing the concentration of $CO_2$ around Rubisco, growth in elevated $CO_2$ concentrations also suppresses photorespiration[43], a process that releases previously fixed $CO_2$. Given that high temperatures stimulate photorespiration[5,9,44], plants grown and measured under elevated $CO_2$ have a higher $T_{optA}$ than those grown and measured at current $CO_2$ levels[9,18,23,30], reflecting the suppression of photorespiration at high temperatures by elevated $CO_2$[9,30,45].

Studies of the thermal sensitivity of photosynthesis have focused on ambient $CO_2$-grown plants[6,8,11,17,20,46], and less on how elevated $CO_2$ may alter temperature acclimation[5,47]. Because of this, the temperature sensitivity functions currently employed in TBMs are derived from ambient $CO_2$-grown plants[6,48]. To date, only a handful of studies have assessed the effect of elevated $CO_2$ on thermal acclimation of photosynthesis[23,30], and only one has investigated the effect of elevated [$CO_2$] on the temperature sensitivity parameters of net photosynthesis and its underlying biochemical processes (maximum Rubisco carboxylation rate–$V_{cmax}$, and maximum electron transport rates–$J_{max}$)[30]. This latter study, conducted on boreal conifer seedlings grown in pots for six months, reported that elevated $CO_2$ had little effect on thermal acclimation of the temperature sensitivity parameters of $V_{cmax}$ and $J_{max}$ (i.e., their thermal optima and activation energies)[30]. In the same study, warming increased $T_{optA}$ by 0.36–0.65 °C per 1 °C warming regardless of $CO_2$ treatments. But elevated $CO_2$-grown seedlings had a $T_{optA}$ that was generally 3.6–4 °C higher than their ambient $CO_2$-grown counterparts when measured at prevailing growth $CO_2$, likely due to direct suppression of photorespiration by elevated $CO_2$.

The key photosynthetic temperature sensitivity parameters employed in TBMs include $T_{optA}$, as well as the thermal optima ($T_{optV}$ and $T_{optJ}$) and activation energies ($E_{aV}$ and $E_{aJ}$) of $V_{cmax}$ and $J_{max}$[6,7]. The responses of these parameters to long-term changes in temperature, either due to experimental warming or natural seasonal variation, are primarily driven by thermal acclimation and less influenced by adaptation to different thermal environments[8,21]. This implies that results generated in this study, using boreal tree species, could have implications for plants grown in natural conditions from different thermal environments.

In this study, we assessed the thermal acclimation of photosynthesis and its underlying biochemical processes (i.e., $V_{cmax}$ and $J_{max}$) in mature trees (~45 years) of tamarack (also known as larch), a deciduous conifer, and black spruce, an evergreen conifer, exposed to either ambient (hereafter aCO$_2$) or elevated $CO_2$ (≈+460 ppm above ambient; hereafter eCO$_2$) combined with a warming of up to +9 °C above ambient temperatures in a regression-based design with five temperature treatments (ambient +0, +2.25, +4.5, +6.75, and +9). The data presented were collected after 2 years of warming combined with one year of $CO_2$ treatment at the Oak Ridge National Laboratory's SPRUCE (Spruce and Peatland Responses Under Changing Environments; https://mnspruce.ornl.gov) project site at the U.S. Forest Service's Marcell Experimental Forest, in Minnesota, USA (47°30.476′ N; 93°27.162′ W).

Here we show that $T_{optA}$ of mature boreal conifers increased with warming, and this warming-induced increases in $T_{optA}$ were correlated with simultaneous increases of the thermal optima of underlying photosynthetic biochemical processes ($V_{cmax}$ and $J_{max}$). However, shifts in $T_{optA}$ did not keep pace with warming as $T_{optA}$ only increased by 0.26–0.35 °C per 1 °C of warming. But when estimated at the mean growth temperature, net photosynthetic rates increased with warming in eCO$_2$ spruce, while remaining constant in aCO$_2$ spruce and in both aCO$_2$ and eCO$_2$ tamarack with warming. Our overall finding is that, although shifts in $T_{optA}$ of these two species are insufficient to keep pace with warming, these boreal conifers can thermally acclimate photosynthesis to maintain carbon uptake in future air temperatures.

## Results

### Shifts in thermal optimum of net photosynthesis ($T_{optA}$)

The $T_{optA}$ increased by 0.26 and 0.35 °C per 1 °C warming in tamarack and black spruce, respectively, and this shift was similar for both aCO$_2$- and eCO$_2$-grown trees (Fig. 1, Supplementary Figs. 1 and 2, and Supplementary Table 1). In addition, $T_{optA}$ was 3 °C higher in eCO$_2$-grown than ambient-grown tamarack, while $CO_2$ had no effect on $T_{optA}$ in black spruce (Fig. 1 and Supplementary Table 1). Warming-induced

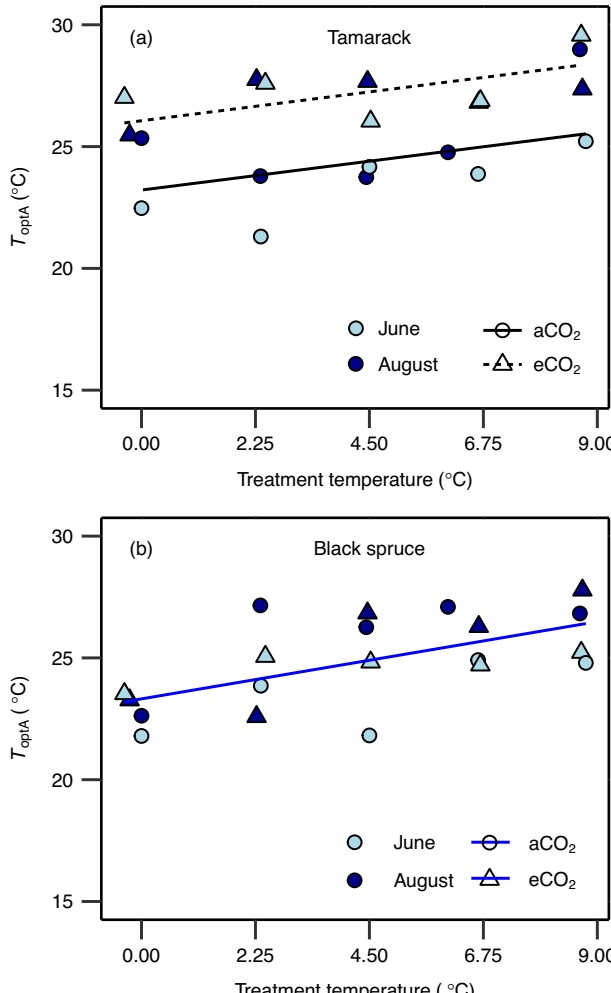

**Fig. 1 | Optimum temperature of net photosynthesis across warming and elevated $CO_2$ treatments.** Impact of temperature and $CO_2$ treatments on the thermal optimum of net photosynthesis ($T_{optA}$, °C) in tamarack (**a**) and black spruce (**b**). The $T_{optA}$ was estimated from temperature response of net photosynthesis measured at growth $CO_2$ using Eq. 2 (see "Methods"). Symbol colors represent the month in which measurements were taken (June = light blue; August = dark blue). Symbol shapes represent $CO_2$ treatments (circle = ambient $CO_2$ – $aCO_2$; triangles = elevated $CO_2$ – $eCO_2$). A mixed-effects regression model was used to analyze the data where warming and elevated $CO_2$ treatment were the fixed effects, and the month in which the campaign was done was the random effect. The statistical test was one-sided since it was done to evaluate whether warming and elevated $CO_2$ increase $T_{optA}$. Lines represent regression lines: in (**a**) the solid ($y = 0.26x + 23.2$; $p = 0.021$) and the short-dashed ($y = 0.26x + 26$; $p = 0.021$) lines represent ambient and elevated $CO_2$ treatments, respectively; in (**b**) the blue line represents the overall regression line when there is no effect of $CO_2$ on the slope and intercept ($y = 0.35x + 23.3$; $p = 0.0058$). Each data point represents the mean value of biologically independent trees measured in each plot ($n = 1$–$4$ trees/plot). Significance threshold: $p < 0.05$. Further details on statistical analyses for this figure can be found in Supplementary Table 1.

increases in $T_{optA}$ were correlated with increases of the thermal optima of photosynthetic biochemical processes, $T_{optV}$ (0.35 and 0.44 °C per 1 °C warming for tamarack and black spruce, respectively) and $T_{optJ}$ (0.26 and 0.55 °C per 1 °C warming for tamarack and black spruce, respectively) (Fig. 2, Supplementary Figs. 3–7, and Supplementary Tables 1 and 2). There was no evidence of acclimation of the activation energy for $V_{cmax}$ in either species (Supplementary Fig. 3e, f). However, in black spruce the activation energy of $J_{max}$ declined non-linearly with warming in $eCO_2$-grown trees but not in $aCO_2$-grown counterparts,

while in tamarack it was unaffected by warming (Supplementary Figs. 3g, h and Supplementary Tables 1 and 2). Furthermore, neither stomatal conductance nor respiration were correlated with the shifts in $T_{optA}$ seen in either species (Supplementary Table 3a, b).

## Exceedance of $T_{optA}$ by mean growth temperature
Photosynthesis typically acclimates to prolonged exposure to warming within 10 days[19–21]. Therefore, we assessed to what extent warming-induced shifts in $T_{optA}$ matched the increases in growth temperature (expressed as the difference between mean air temperature for the 10 days preceding each measurement and the respective $T_{optA}$; $\Delta MeanT_g$). This approach assumes that leaf and air temperatures are similar, a reasonable assumption considering the tight coupling between leaf and air temperature in small leaves[49], such as conifer needles. In $aCO_2$-grown tamarack and black spruce, mean daytime growth temperature exceeded $T_{optA}$ ($\Delta MeanT_g > 2$ °C) across all warming treatments (+2.25 to +9 °C) (Fig. 3 and Supplementary Table 4). However, $eCO_2$ reduced the $\Delta MeanT_g$ for tamarack in the +2.25 °C treatment, while for black spruce, $eCO_2$ had weak or no effect on $\Delta MeanT_g$ across all warming treatments (Fig. 3b, d and Supplementary Table 4).

## Elevated $CO_2$ impacts on thermal sensitivity of net photosynthesis
We also examined the impact of the treatments on the model parameter representing the spread of the instantaneous temperature response of net photosynthesis ($b$ in Eq. 2, see "Methods"). A high $b$ value represents a narrower temperature response curve of photosynthesis and thus higher sensitivity to short-term temperature fluctuations[21]. In both species, $b$ was unaffected by warming in $aCO_2$-grown trees. However, the impact of $eCO_2$ differed between the two species. In tamarack, $b$ was constant in $eCO_2$-grown trees across the warming treatments, but 86% higher than in the $aCO_2$ tamarack (Supplementary Fig. 9 and Supplementary Table 1), suggesting an overall $CO_2$-induced increase in short-term temperature sensitivity (Supplementary Fig. 2). In contrast, in black spruce, $CO_2$ had no effect on $b$ in the temperature control treatments (+0). However, $b$ marginally increased ($p = 0.067$) with warming in the $eCO_2$-grown trees, such that it was 68% higher in $eCO_2$ than in AC in the warmest plot (+9 °C) (Supplementary Fig. 9 and Supplementary Table 1), suggesting an $eCO_2$-induced increase in the temperature sensitivity of net photosynthesis as it gets warmer (Supplementary Fig. 3).

## Net photosynthetic rates at the $T_{optA}$ and growth temperature
Thermal acclimation of net photosynthesis can also be assessed by examining the extent to which net photosynthetic rates at the thermal optimum ($A_{opt}$) and at prevailing growth temperature are affected by warming[17]. In tamarack, $A_{opt}$ was constant across the warming treatments but with overall higher rates in $eCO_2$ trees compared to their $aCO_2$ counterparts (Fig. 4a and Supplementary Table 1). By contrast, in black spruce, there was an interaction of warming and elevated $CO_2$ such that $A_{opt}$ significantly increased with warming in $eCO_2$ trees, while it was constant across warming in $aCO_2$ trees (Fig. 4b and Supplementary Table 1). Moreover, net photosynthetic rates estimated at mean ($A_g$) growth temperature exhibited similar responses to $A_{opt}$ in both species (Fig. 5 and Supplementary Table 1). These results suggest that, overall, the two species were able to maintain their carbon uptake at prevailing growth temperatures. We further estimated net $CO_2$ assimilation at growth temperature conditions for 2 years (2016 and 2017), representing the entire acclimation period to temperature in this study. The results show that net $CO_2$ assimilation rates were not negatively affected by warming in either species throughout the growth seasons of both 2016 and 2017. In tamarack, $A_g$ was constant across warming and $CO_2$ treatments throughout the growth seasons of the 2 years (Supplementary Fig. 10 and Supplementary Table 5).

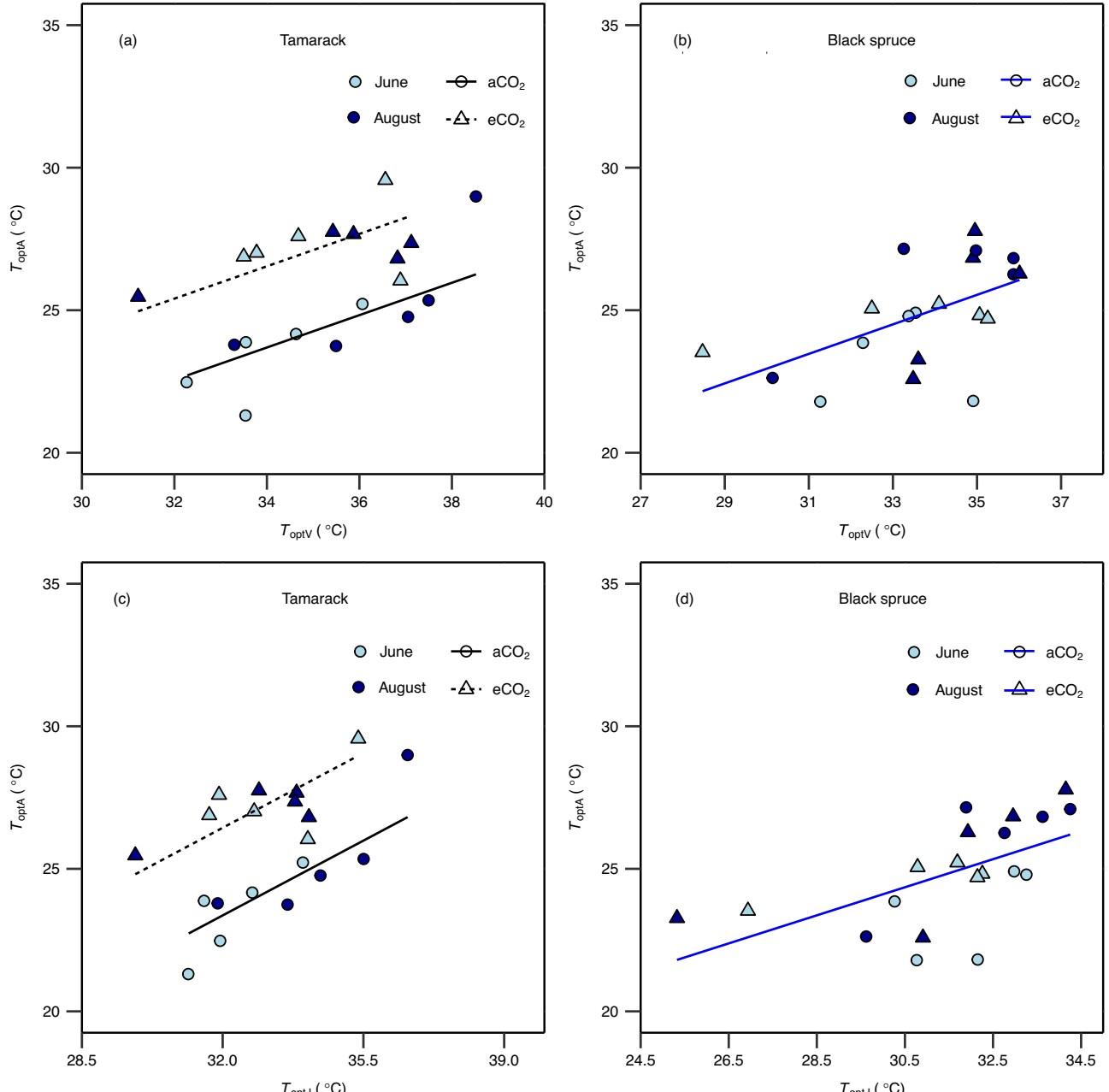

**Fig. 2 | Relationship between the optimum temperature of net photosynthesis and the optima temperatures of underlying biochemical processes.** The temperature optimum of net photosynthesis measured at growth $CO_2$ ($T_{optA}$, °C) as a function of the thermal optimum of **a, b** the maximum Rubisco carboxylation rate ($T_{optV}$, °C); **c, d** the maximum electron transport rate ($T_{optJ}$, °C) in tamarack (**a, c**) and black spruce (**b, d**). Symbol colors represent the month in which measurements were taken (June = light blue; August = dark blue). Symbol shapes represent $CO_2$ treatments (circle = ambient $CO_2$−a$CO_2$; triangles = elevated $CO_2$−e$CO_2$). A mixed-effects regression model was used to analyze the data where warming and elevated $CO_2$ treatment were the fixed effects, and the month in which the campaign was done was the random effect. The statistical test was one-sided since it was done to evaluate whether there is a positive relationship among the thermal optima of net photosynthesis and underlying biochemical processes. Lines represent regression lines: in (**a, c**) the solid (**a**: $y = 0.57x + 4.4$, $p = 0.0011$; **c**: $y = 0.75x − 0.62$, $p < 0.0001$) and short-dashed (**a**: $y = 0.57x + 7.1$, $p = 0.0011$; **c**: $y = 0.75x + 2.4$, $p < 0.0001$) lines represent ambient and elevated $CO_2$ treatments, respectively; in (**b, d**) the blue line (**b**: $y = 0.52x + 7.4$, $p = 0.0108$; **d**: $y = 0.56x + 7.3$, $p = 0.0026$) represents overall regression line when there is no effect of $CO_2$ on the slope and intercept. Each data point represents the mean value of biologically independent trees measured in each plot ($n = 1$–4 trees/plot). Significance threshold: $p < 0.05$. Further details on statistical analyses for this figure can be found in Supplementary Table 2.

In black spruce, $A_g$ was largely constant across warming treatments in both years for a$CO_2$ trees, while for e$CO_2$ trees, $A_g$ commonly increased with warming (Supplementary Fig. 11 and Supplementary Table 5).

## Discussion

We report findings, to our knowledge, from the first field study assessing responses of the short-term temperature sensitivity of photosynthesis to long-term exposure to whole-ecosystem warming (2 years) combined with elevated atmospheric $CO_2$ (1 year) in mature trees (~45 years old). These results provide a benchmark for our understanding of the impacts of these climate change variables (and their potential interaction) on the thermal sensitivity of photosynthesis in long-lived trees that are experiencing gradual increases in temperature and atmospheric $CO_2$ in their natural environment.

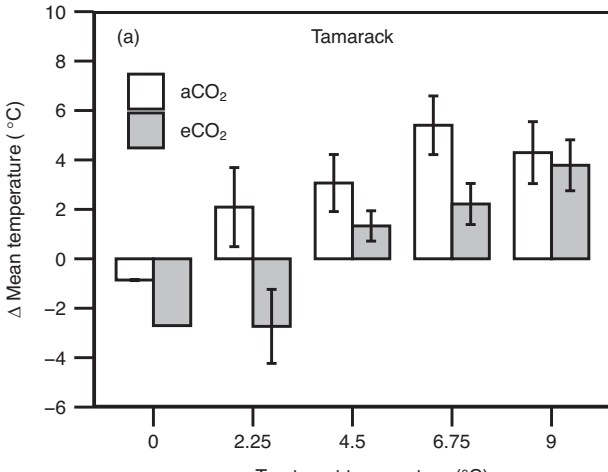

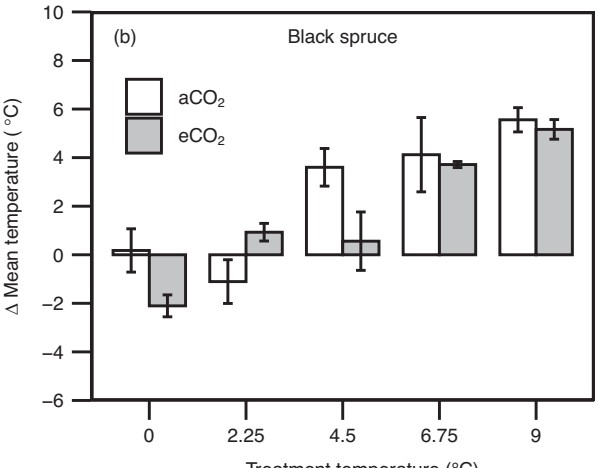

**Fig. 3 | Changes in the difference between the optimum temperature of net photosynthesis and prevailing growth temperature across warming and elevated CO₂ treatments.** Difference (Δ) between mean daytime (9 am to 3 pm–time of the day when plants are most photosynthetically active) air temperature (°C) and the temperature optimum of net photosynthesis measured at growth CO₂ ($T_{optA}$, °C) for tamarack (**a**) and black spruce (**b**). The mean daytime air temperature corresponded to the average temperature across 10 days prior to each measurement day. Bar colors represent CO₂ treatment (white = ambient CO₂–aCO₂; gray = elevated CO₂–eCO₂). For (**a**), $n$ = 2, 1, 3, 3, 2, 3, 3, 3, 3, and 3 biologically independent trees for +0 aCO₂, +0 eCO₂, +2.25 aCO₂, +2.25 eCO₂, +4.5 aCO₂, +4.5 eCO₂, +6.75 aCO₂, +6.75 eCO₂, +9 aCO₂, and +9 eCO₂ treatments, respectively; for (**b**), $n$ = 2, 3, 3, 3, 3, 2, 3, 3, 3, and 3 biologically independent trees for +0 aCO₂, +0 eCO₂, +2.25 aCO₂, +2.25 eCO₂, +4.5 aCO₂, +4.5 eCO₂, +6.75 aCO₂, +6.75 eCO₂, +9 aCO₂, and +9 eCO₂ treatments, respectively. Mean ± SE. Further details on statistical analyses for this figure can be found in Supplementary Table 4.

We show that the thermal optimum of net photosynthesis ($T_{optA}$) increased by 0.26–0.35 °C per °C warming in mature boreal conifers (Fig. 1). These results are comparable to those from a long-term (3 years) field-based warming study with boreal and temperate seedlings, which reported a rise in $T_{optA}$ of -0.38 °C per °C warming[19]. However, our $T_{optA}$ values were largely exceeded by mean daytime growth temperature under current atmospheric CO₂ conditions (Fig. 3), suggesting that shifts in $T_{optA}$ in mature trees of boreal conifers growing in the natural field conditions may not fully adjust to compensate for increases in ambient air temperatures. Therefore, exceedance of $T_{optA}$ by prevailing mean air temperatures across treatments implies that the frequent and severe heat stress predicted under climate change will further constrain carbon uptake in boreal forest conifers.

Until now, knowledge of the thermal acclimation of $T_{optA}$ and its underlying processes was largely based on short-term studies with seedlings grown in artificial growth environments (e.g., pots) and in controlled environmental conditions (e.g., humidity, light). It was thus unclear whether those results would hold for mature trees growing in the field. Observed shifts in $T_{optA}$ in our study are at the lower end of the spectrum (0.35–0.8 °C per 1 °C) reported for lab-based experimental studies with seedlings[22, 30,50], but are comparable to mean values reported by recent meta-analyses for C₃ plants (0.34[31] and 0.38[11] °C per 1 °C), indicating that while seedlings may have a greater ability to acclimate photosynthesis to warming than mature trees, average responses of photosynthetic thermal acclimation can be broadly used. Furthermore, the shift in $T_{optA}$ with warming in our field study is much lower than that from a recent global compilation (0.62 °C per 1 °C) that estimated shifts in $T_{optA}$ using seasonal changes in temperature (i.e., acclimatization). Therefore, we suggest that the use of temperature sensitivity parameters derived from 'acclimatization studies' should be used with caution when predicting the acclimation of forests to warming in global vegetation models. We also show that thermal acclimation of $T_{optA}$ is strongly driven by concomitant adjustments of the thermal optima of photosynthetic biochemical processes (Fig. 2), and not changes in stomatal conductance or respiration (Supplementary Table 3a, b), findings that agree with prior work on controlled experiments in seedlings[29,30,51], field warming experiments[19,21,52], and a recent acclimatization study[8]. These results imply that changes in photosynthetic biochemical processes strongly underlie the adjustment of photosynthesis to long-term changes in growth temperature, regardless of experimental approach or tree life stage, although stomatal limitations are likely to play a greater role in limiting photosynthesis in water-stressed trees.

Most studies that have examined thermal acclimation of photosynthesis did so on ambient CO₂-grown trees. Elevated CO₂ is expected to influence the thermal acclimation of photosynthetic biochemistry (i.e., maximum Rubisco carboxylation rate, $V_{cmax}$, and maximum electron transport rates, $J_{max}$) mainly due to its suppressive effect on photorespiration[53,54] and its direct effects on Rubisco carboxylation[35–37], both of which are temperature dependent processes[5]. However, we show that elevated CO₂ does not largely affect the thermal optima or activation energies of $V_{cmax}$ or $J_{max}$ (Supplementary Fig. 3 and Supplementary Table 1). These findings with field-grown mature boreal trees agree with an earlier, short-term study with seedlings of the same species[30], suggesting that regardless of the experimental approach, life stage, and leaf habit, elevated CO₂ does not have strong effects on the thermal sensitivity of photosynthetic biochemical processes, such as $V_{cmax}$ and $J_{max}$, in boreal conifers. Since $V_{cmax}$ and $J_{max}$ are key parameters for representing carbon uptake within TBMs[6], our findings imply that potential interactive effects of elevated CO₂ on temperature sensitivity parameters of $V_{cmax}$ and $J_{max}$ (i.e., their activation energies and thermal optima) can be ignored in TBMs. Our findings also suggest that temperature response functions of these parameters, developed mainly from ambient CO₂-grown plants[8,46] and currently employed in all TBMs[6,7,48] might accurately represent carbon uptake for trees growing in both current and projected elevated CO₂ conditions in future climates. However, further research on tree species from other biomes and plant functional types (e.g., broadleaved tree and shrub species) are still needed to validate this conclusion for broad use.

We show that the *b* parameter was generally increased by elevated CO₂ for both species, suggesting that elevated CO₂ increases the thermal sensitivity of net photosynthesis, a result in line with a shift to photosynthesis being more RuBP-regeneration limited at high CO₂ concentrations[9]. In addition, elevated CO₂ did affect the $T_{optA}$, but these effects were species dependent. In tamarack, the $T_{optA}$ was higher in elevated CO₂, which likely reflects a direct suppression of photorespiration[5,9,43,54]. In contrast, there was no effect of elevated CO₂ on the $T_{optA}$ in black spruce, and these results contrast prior findings in

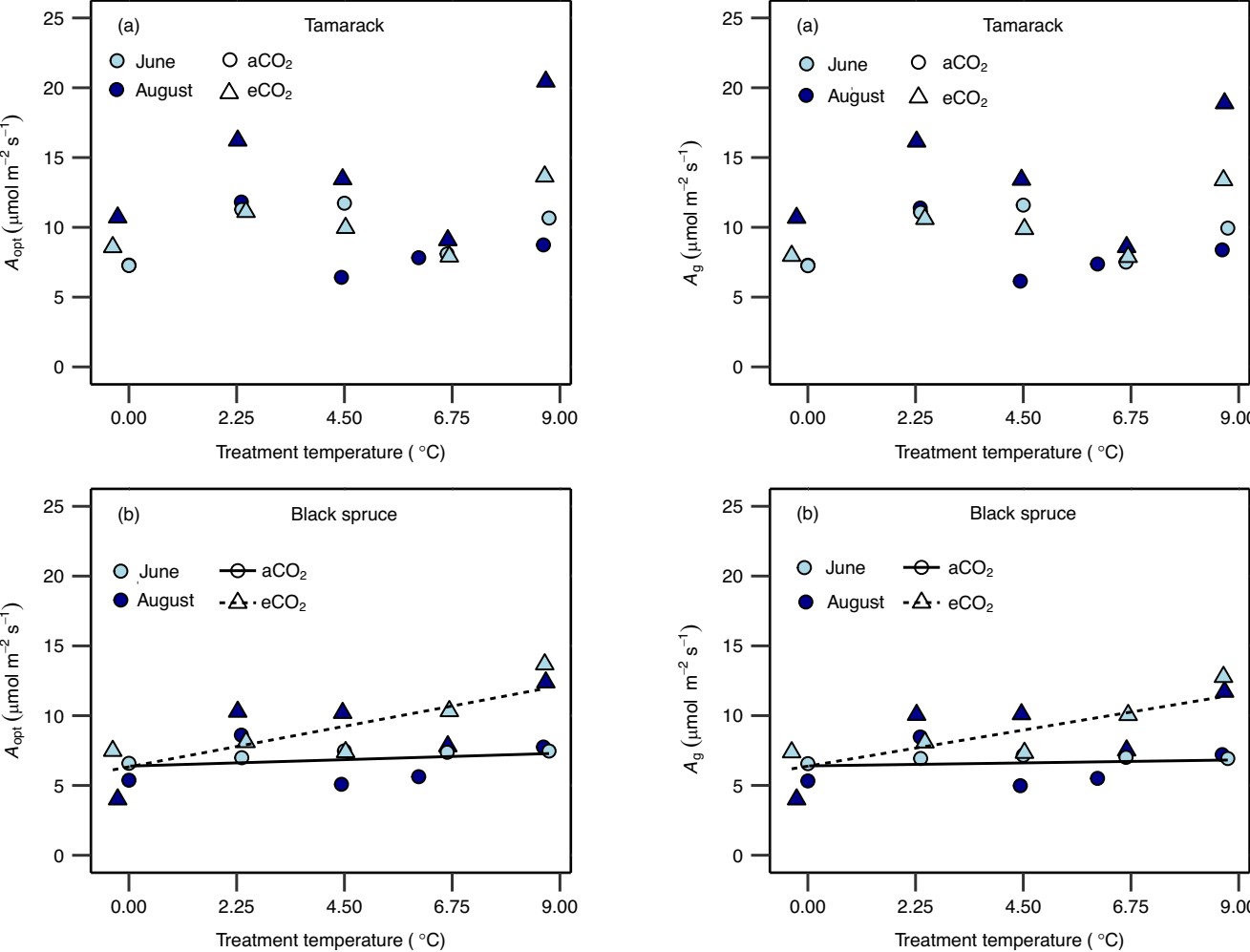

**Fig. 4 | Net photosynthetic rates at the thermal optimum across warming and elevated $CO_2$ treatments.** Impact of temperature and $CO_2$ treatments on net photosynthesis rate at the thermal optimum temperature ($A_{opt}$) in tamarack (**a**) and black spruce (**b**). The $A_{opt}$ was estimated from temperature response of net photosynthesis measured at growth $CO_2$ using Eq. 2 (see "Methods"). Symbol colors represent the month in which measurements were taken (June = light blue; August = dark blue). Symbol shapes represent $CO_2$ treatments (circle = ambient $CO_2$–a$CO_2$; triangles = elevated $CO_2$–e$CO_2$). A mixed-effects regression model was used to analyze the data where warming and elevated $CO_2$ treatment were the fixed effects, and the month in which the campaign was done was the random effect. The statistical test was one-sided since it was done to evaluate whether warming and elevated $CO_2$ stimulate $A_{opt}$. Lines in (**b**) represent regression lines: the solid ($y = 0.10x + 6.4$; $p = 0.54$) and the short-dashed ($y = 0.54x + 6.3$; $p = 0.029$) lines represent ambient and elevated $CO_2$, respectively. In (**a**), $A_{opt}$ did not significantly change with treatments ($y = 0.26x + 7.9$, $p = 0.27$ and $y = 0.26x + 10.89$, $p = 0.27$, for ambient and elevated $CO_2$ treatments, respectively). Each data point represents the mean value of biologically independent trees measured in each plot ($n = 1–4$ trees/plot). Significance threshold: $p < 0.05$. Further details on statistical analyses for this figure can be found in Supplementary Table 1.

**Fig. 5 | Net photosynthetic rates at prevailing growth temperatures across warming and elevated $CO_2$ treatments.** Impact of temperature and $CO_2$ treatments on net photosynthesis rate estimated at mean growth temperature (9 a.m.–3 p.m.; $A_g$) in tamarack (**a**) and black spruce (**b**). Symbol colors represent the month in which measurements were taken (June = light blue; August = dark blue). Symbol shapes represent $CO_2$ treatments (circle = ambient $CO_2$–a$CO_2$; triangles = elevated $CO_2$– e$CO_2$). A mixed-effects regression model was used to analyze the data where warming and elevated $CO_2$ treatment were the fixed effects, and the month in which the campaign was done was the random effect. The statistical test was one-sided since it was done to evaluate whether warming and elevated $CO_2$ stimulate $A_g$. Lines in (**b**) represent regression lines: the solid ($y = 0.047x + 6.4$; $p = 0.76$) and the short-dashed ($y = 0.53x + 6.4$; $p = 0.026$) lines represent ambient and elevated $CO_2$, respectively. In (**a**), $A_g$ did not significantly change with treatments ($y = 0.2x + 7.9$, $p = 0.35$ and $y = 0.2x + 10.79$, $p = 0.35$, for ambient and elevated $CO_2$ treatments, respectively). Each data point represents the mean value of biologically independent trees measured in each plot ($n = 1–4$ trees/plot). Significance threshold: $p < 0.05$. Further details on statistical analyses for this figure can be found in Supplementary Table 1.

black spruce seedlings[30]. The reasons behind this lack of elevated $CO_2$ effect on $T_{optA}$ in mature black spruce are unclear since, similar to tamarack, the needle cohorts that were measured developed in prevailing environmental conditions across treatments. However, the magnitude of suppression of photorespiration by elevated $CO_2$ may vary across species or plant functional types—or in our case differences in leaf habit (evergreen versus deciduous). In our study, we cannot make a solid conclusion on the main cause for this, but two possibilities are differences in stomatal (Supplementary Fig. 13 and

Supplementary Table 1) and mesophyll conductance between the species. Our data shows little differences in intracellular $CO_2$ concentration between the two species across the two $CO_2$ treatments (Supplementary Fig. 14 and Supplementary Table 1), indicating that stomatal limitations are unlikely to underlie the difference in how $T_{optA}$ responds to elevated $CO_2$. This leaves mesophyll conductance as a possible factor, as higher mesophyll conductance in tamarack could enhance $CO_2$ supply to Rubisco for a given unit of intercellular $CO_2$. However, without mesophyll conductance measurements we cannot directly prove this, and future research is needed to investigate this possibility.

Even though prevailing air temperatures largely exceeded $T_{optA}$, our findings show that photosynthesis acclimated such that at the prevailing daytime mean air temperature (between 9 a.m. and 3 p.m.), net carbon fixation remained constant or even increased (in $eCO_2$ black spruce trees) across the warming treatments (Fig. 5 and Supplementary Figs. 10 and 11). Therefore, our findings imply that warming alone may have little negative impacts on leaf-level carbon uptake in these cold-adapted mature boreal conifers when soil moisture is not limiting[55], as is the case at our current study site[56]. However, ongoing climate change and the increased frequency of strong heat and dry spell events that will accompany it will likely reduce the ability of forests dominated by these species to fix and sequester carbon[57]. Moreover, increased autotrophic respiration, which is temperature-dependent, has also been indicated as another factor that will release carbon sequestered in these North American boreal forests[58]. Our previous work from this experiment support this, where we showed that foliar dark respiration did not thermally acclimate in these boreal conifers[56], suggesting that although carbon fixation may not be negatively impacted by warming, thermal effects on autotrophic respiration will further reduce the carbon sequestration potential of these forests[59].

In summary, our study has implications for the understanding of climate warming effects on carbon uptake of mature boreal conifers growing in field conditions, and for improving the representation of photosynthesis in TBMs. First, we show that although thermal acclimation of $T_{optA}$ is limited and does not fully match increases in air temperature, photosynthetic carbon fixation is maintained at the prevailing growth conditions through a combination of photosynthetic acclimation and changes in instantaneous temperature responses of photosynthetic processes. Second, our study provides an improved framework for modeling photosynthesis in TBMs considering both warming and elevated $CO_2$, because we provide support for ignoring effects of elevated $CO_2$ on the thermal sensitivity of photosynthetic biochemical parameters (thermal optima and activation energies of $V_{cmax}$ and $J_{max}$)[22]. However, we show that it is important to account for effects of elevated $CO_2$ on the $T_{optA}$ and on the overall thermal sensitivity of net photosynthesis ($b$ parameter).

## Methods

### Site description and experimental design
This study was conducted at the Oak Ridge National Laboratory's SPRUCE (Spruce and Peatland Responses Under Changing Environments) project site at the U.S. Forest Service's Marcell Experimental Forest, in Minnesota, USA (47°30.476′ N; 93°27.162′ W). The details of the study site and experimental design are provided in recent studies from this experiment[56,60–62]. But briefly, this forest grows naturally in a bog located at the southern limit of the boreal peatland forests. The forest is approximately 50 years old as it regenerated following canopy tree removal in 1969 and 1974[63]. The dominant canopy species is *Picea mariana* (Mill.) B.S.P. (black spruce) mixed with less abundant *Larix laricina* (Du Roi) K. Koch (tamarack). The understory vegetation is dominated by ericaceous shrubs *Rhododendron groenlandicum* (Oeder) Kron & Judd and *Chamaedaphne calyculata* (L.) Moench. The experiment comprises five temperature treatments (ambient or +0, which serves also as the control, +2.25, +4.5, +6.75, and +9 °C above the ambient) established in a regression-based design[64]. This experiment uses 10 large octagonal open-top enclosures with an interior surface area of 114.8 m², and a sampling area of 66.4 m². Five enclosures have an ambient-$CO_2$ atmosphere, while the other five have an elevated $CO_2$ atmosphere varying between +430 and 500 ppm above the ambient. The heating treatments started August 15, 2015, and $CO_2$ treatments were initiated a year later, on June 15, 2016. The targeted temperature treatments and $CO_2$ concentrations were largely achieved (Supplementary Fig. 12).

### Plant material sampling and gas exchange measurements
Field measurements were conducted between June 18–30 and August 15–30, 2017. The daytime temperatures (4:00 a.m.–8:30 p.m.) during June and August were 18.97 and 18.02 °C, respectively. We studied the two mixed-age (up to ~45 years old) canopy tree species at SPRUCE, *Picea mariana* (Mill.) B.S.P. (black spruce) and *Larix laricina* (Du Roi) K. Koch (tamarack). For black spruce, one branchlet for each, randomly selected tree and in each plot was harvested and 1-year needle cohorts (i.e., developed in growth season of 2016) from each branch was measured. For tamarack, fully expanded current year foliage was used. In the June field campaign, three trees in each plot were randomly sampled, while in the August campaign, only two trees were used. For tamarack, we used the same number of branchlets from different trees in each plot, except in one plot (in ambient $CO_2$ and +0) where only one tamarack tree was available to be sampled. All measurements were made on sun-exposed branchlets cut using a pruning pole. After cutting, branchlets were put in water, and recut under water to avoid xylem transport disruption and stomatal closure. The branches were harvested between 4 and 5 a.m. of the measurement day, placed in water bottles inside a plastic cooler, and transported from the field site in Marcell, Minnesota to the walk-in growth chambers at the University of Minnesota in St. Paul, where the measurements were conducted. The branchlets were re-cut again before starting the measurements. The effect of cutting and the time lag between cutting and gas exchange measurements has been shown not to have significant effect on stomatal conductance in conifers[65]. Gas exchange measurements were conducted between 10:00 and 20:00 using 7 portable photosynthesis systems (Li-COR 6400 XT, 6400-18 RGB light source, and 6400-22 opaque conifer chamber; LI-COR Biosciences, Lincoln, NE, USA). Net $CO_2$ assimilation rates ($A$) were measured at a predetermined saturating light (1800 μmol m⁻² s⁻¹) and eleven different air $CO_2$ concentrations (to generate so-called $A$–$C_i$ curves) in the following order: 400, 300, 200, 50, 400, 500, 600, 800, 1200, 1600, and 2000 μmol mol⁻¹. The $A$–$C_i$ curves were conducted at five different leaf temperatures ($T_{leaf}$): 15, 25, 32.5, 40, and 45 °C. In order to achieve each targeted $T_{leaf}$, all measurements were completed inside the growth chamber, allowing the entire branch to be exposed to the desired temperature for at least 30 min before starting measurements at that temperature. As the gas exchange systems were also inside the chamber, this method minimized the measurement error driven by the internal thermal gradient that was recently reported for the LI-6400 instruments[66]. Since the vapor pressure of the air ($VPD_{air}$) increases with increasing temperature, resulting in decreased stomatal conductance[15], we moistened the soda lime column of the gas exchange systems to reduce stomatal closure associated with high VPDs at high measurement temperatures (>32.5 °C). In total, we present results of 96 $A$–$C_i$ temperature response curves. After gas exchange measurements, projected leaf area of the measured needles was determined using ImageJ 1.51 software (NH, Bethesda, MD, USA). We, thereafter, corrected for the total leaf area before the analyses.

### Parameterization
The FvCB (Farquhar, von Caemmerer, and Berry) $C_3$ photosynthesis model[67] was used to derive $V_{cmax}$ and $J_{max}$ from the $A$–$C_i$ curves using the fitacis function from the plantecophys 1.4-6 R package[68] and using the bilinear fitting method. We maintained the default temperature dependencies of the $CO_2$ compensation point in the absence of mitochondrial respiration ($\Gamma^*$) and the Michaelis–Menten constants for $CO_2$ and $O_2$ ($K_c$ and $K_o$) from Bernacchi et al.[69]. The leaf mesophyll conductance for $CO_2$ was not measured, therefore apparent $V_{cmax}$ and $J_{max}$ based on intercellular $CO_2$ concentrations ($C_i$), rather than the $CO_2$ concentration at the site of carboxylation ($C_c$) in the chloroplast, were estimated. The temperature sensitivity parameters of $V_{cmax}$ ($T_{optV}$ and $E_{aV}$) and $J_{max}$ ($T_{optJ}$ and $E_{aJ}$) were derived using the modified Arrhenius

function outlined in the following Eq. 1[70]:

$$f\left(T_k\right) = k_{opt} \frac{H_d \exp\left(\frac{E_a\left(T_k - T_{opt}\right)}{T_k R T_{opt}}\right)}{H_d - E_a\left(1 - \exp\left(\frac{H_d\left(T_k - T_{opt}\right)}{T_k R T_{opt}}\right)\right)} \tag{1}$$

where $k_{opt}$ is the process rate (i.e., $V_{cmax}$ or $J_{max}$; $\mu mol\,m^{-2}\,s^{-1}$) at the optimum temperature ($V_{cmaxopt}$, $J_{maxopt}$), $H_d$ (kJ $mol^{-1}$) is the deactivation energy term that describes the decline in enzyme activity at higher temperature, $E_a$ (kJ $mol^{-1}$) is the activation energy term that describes the exponential increase in enzyme activity with an increase in temperature, $R$ is the universal gas constant (8.314 J $mol^{-1}\,K^{-1}$), and $T_{opt}$ and $T_k$ are the optimum and given temperatures of the process rate (i.e., $V_{cmax}$ or $J_{max}$; $\mu mol\,m^{-2}\,s^{-1}$). The value of $H_d$ was fixed at 200 kJ $mol^{-1}$ to avoid over-parameterization[70,71].

Net photosynthesis data at the tree growth $CO_2$ (400 or ~800 ppm, for ambient $CO_2$ and elevated $CO_2$ treatments, respectively) were extracted from the $A$–$C_i$ curves. Thereafter, the temperature response of $A$ was fitted using the following Eq. 2[16] to estimate the $T_{optA}$:

$$A(T) = A_{opt} - b\left(T - T_{optA}\right)^2 \tag{2}$$

where $A$ ($T$) is the $A$ ($\mu mol\,m^{-2}\,s^{-1}$) at a given air temperature $T$ (°C), $A_{opt}$ is the $A$ at the optimum temperature ($T_{opt}$), and the $b$ parameter represents the breadth of the photosynthetic temperature response curve; larger values of $b$ indicates that $A$ ($T$) has greater sensitivity to changes in $T$. After fitting $b$ and $T_{optA}$, we used Eq. 2 to model net photosynthesis at prevailing growth temperature conditions using mean and maximum air temperature (9–4 a.m.) for each plot for 10 days preceding measurement of each tree/species, as well as for the entire growing season period (June–September) of both 2016 and 2017.

In order to estimate to what extent stomatal conductance may have affected the shifts in $T_{optA}$, we re-calculated net photosynthesis at a $C_i/C_a$ ratio of 0.7 ($A_{70}$; with a final $C_i$ of 280 or 560 ppm for ambient and elevated $CO_2$, respectively) using the parameterized $V_{cmax}$, $J_{max}$, $R_{day}$, and TPU (triose phosphate use) from the plantecophys 1.4-6 R package in the following equations[44]:

$$A_c = \frac{V_{cmax}(C_i - \Gamma^*)}{\left[C_i + K_c\left(1 + \frac{O}{K_O}\right)\right]} - R_{day} \tag{3}$$

where O the intercellular concentrations of $O_2$, $K_c$ and $K_O$ are the Michaelis–Menten coefficients of Rubisco activity for $CO_2$ and $O_2$, respectively, and $\Gamma^*$ is the $CO_2$ compensation point in the absence of mitochondrial respiration. Values at 25 °C and temperature sensitivities of $\Gamma^*$, $K_c$ and $K_O$ were taken from Bernacchi et al.[69].

$$A_j = \left(\frac{J_{max}}{4}\right) \times \frac{\left(C_i - \Gamma^*\right)}{\left(C_i + 2\Gamma^*\right)} - R_{day} \tag{4}$$

$$A_{TPU} = 3TPU \tag{5}$$

$A_{70}$ was considered as the minimum of $A_c$, $A_j$, and $A_{TPU}$, and $T_{optA}$ of $A_{70}$ was estimated using Eq. 2.

**Statistical tests.** In order to evaluate the effect of elevated $CO_2$ on the thermal acclimation of the photosynthetic parameters, we used a mixed-effects regression model where warming and elevated $CO_2$ treatment were the fixed effects, and the month in which the campaign was done was the random effect. All analyses were run on the plot means with $n = 1$–4 trees/plot. The selection of the final statistical model was done in two steps following the protocol proposed by Zuur et al.[72]. We first evaluated whether a random factor was required by comparing the model with the random intercept (i.e., month) with the model without any random structure using the gls function of the nlme 3.1.162 R Package[73] and the method set to the Restricted maximum likelihood (REML). We did not include the model with a random slope and intercept structure since preliminary analyses indicated that the statistical model was over-parameterized. Thereafter, the model with the adequate random structure was selected based on the lowest AIC (Akaike Information Criterion) using the R anova function. After, the selection of the adequate random structure, we then selected for the adequate fixed effect structure between the structure with just main effects (i.e., warming and elevated $CO_2$) without interaction and with interaction. The latter selection was done by comparing these two fixed effect structures using the maximum likelihood–ML method within the gls function. Similarly, the best fixed effect structure was selected based on the lowest AIC value. But because our sample size is relatively small, we then computed the AICc using AICmodavg 2.3-2 R package[74] (Supplementary Tables 6 and 7). We also run ANOVA tests to examine the effects of temperature and elevated $CO_2$ treatments on delta-mean temperature growth ($\Delta MeanTg$; Supplementary Table 4). All analyses were conducted in R 3.6.1 software. (R Core Team, 2019), except for unpaired $t$ Tests (Supplementary Table 3a, b) that were performed using statistical package in Excel 16.74 software.

**Reporting summary**
Further information on research design is available in the Nature Portfolio Reporting Summary linked to this article.

## Data availability
The raw and processed (i.e., mean values used to generate each figure in the paper) photosynthesis data generated in this study have been deposited in the figshare database and can be accessed at https://doi.org/10.6084/m9.figshare.22645030[75]. The complete leaf gas exchange data, including the data used in this paper, are also available through the SPRUCE project website at https://doi.org/10.25581/spruce.056/1455138[76].

## Code availability
The R codes used for analyses for each figure included in this paper can be accessed at https://doi.org/10.6084/m9.figshare.22645030[75].

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

## Acknowledgements

Research was sponsored by the Biological and Environmental Research Program in the Office of Science, U.S. Department of Energy managed by UT- Battelle, LLC, for the U.S. Department of Energy under contract DEAC05-00OR22725. M.E.D., J.M.W., E.J.W., D.A.M., A.W.K. and P.J.H. were supported under this contract. E.J.W. also acknowledges support from USGS Climate Research and Development Program. P.B.R., A.S., R.B., and R.A.M acknowledge funding support by the U.S. NSF Biological Integration Institutes grant DBI-2021898. D.A.W. acknowledges funding from the NSERC Discovery and Strategic programs (RGPIN/04677-2019 and STPGP/521445-2018), the Research School of Biology at the Australian National University, and the U.S. Department of Energy contract No. DE-SC0012704 to Brookhaven National Laboratory. Notice: This manuscript has been authored by UT-Battelle, LLC, under contract DE-AC05-00OR22725 with the US Department of Energy (DOE). The US government retains and the publisher, by accepting the article for publication, acknowledges that the US government retains a non-exclusive, paid-up, irrevocable, worldwide license to publish or reproduce the published form of this manuscript, or allow others to do so, for US government purposes. DOE will provide public access to these results of federally sponsored research in accordance with the DOE Public Access Plan (http://energy.gov/downloads/doe-public-access-plan).The DOI link for the dataset used in this paper can be accessed at https://doi.org/10.25581/spruce.056/1455138 and https://doi.org/10.6084/m9.figshare.22645030. Any use of trade, firm, or product names is for descriptive purposes only and does not imply endorsement by the U.S. Government.

## Author contributions

M.E.D., J.M.W., E.J.W., P.J.H., and D.A.W. designed the research; M.E.D., J.M.W., E.J.W., B.K.M., A.S., R.B., M.C., D.A.M., and A.W.K. collected the data; M.E.D. analyzed the data; M.E.D. wrote the manuscript with significant contributions from J.M.W., P.B.R., and D.A.W.; E.J.W., B.K.M., A.S., R.B., M.C., D.A.M., A.W.K., R.A.M., and P.J.H. provided editorial advice. All co-authors commented on versions of the manuscript.

## Competing interests

The authors declare no competing interests.
