## [Peer Review File · Nature Communications]

REVIEWER COMMENTS

Reviewer #1 (Remarks to the Author):

Review: Boreal conifers maintain carbon uptake with warming despite failure to track optimal temperatures

General:

This is a really important study from the SPRUCE experiment. The results are also interesting! For the most part, the writing flowed well and interpretation was easy, but I did get bogged down a few times in the details of the results section. The discussion helped with this. I suggest a few things below to help out with the abstract and to help the reader find the most important points. Finally, while I really liked the clear explanation of TBM implications, I think dismissing the need to develop temperature acclimation in models could use a caveat – this was only for two species. There are a lot of other conifers (globally and in North America) that may respond differently. I know we can't experiment on all of them, but perhaps it should be mentioned that this type of trait plasticity is still unknown for non-boreal species (for field grown mature trees).

Abstract:

1. "but these increases were not sufficient to keep pace with warming" – What does this mean? T_{opt} lagged or just changed less? What are the consequences of this? I think both a quantitative example of what this means (0.X per X degrees C) and what the consequences are of T_{opt} not keeping pace would be more powerful.
2. Can you indicate early on that Tamarack is deciduous (and maybe say Larch as well)? I don't think everyone knows this.
2. I was pretty confused about the 'modeling' statement. Part of this was just because I hadn't worked out what you were trying to point out with mean growth temperatures, but the other reason is because I thought you meant you process-based modeled it (like with a TBM or an ecosystem model). If you could point out that the result is from the equations (yes, they are also process-based, but are not a TBM) that would help the reader understand that there is not a major modeling component to the study (like a TBM exercise).

Results:

1. Lines 154 – 157: had me a bit confused. Both sentences are saying the same thing for both species but as written, one expects them to be different. Can you just say they both did the same thing and reduce the words? (Yes there is a different range of temperatures for the species but it is the same 'most of the time')
2. Tamarack is deciduous (larch?)....does that matter? Deciduous leaves less likely to 'acclimate'? Or would new leaves acclimate? This is explained (one sentence) later but I think it is really interesting that you have these results from a evergreen and deciduous conifer and I would have like a little more discussion about why they may have been expected to respond differently (was there a hypothesis?)

Discussion

3. LINE 205 to 206 is an important point! Maybe put it in the abstract?
4. Lines 239 – 245: Really important point! But, this was only for two species.

Figures.

Figure 1: Can there be a legend (instead of just what the symbols mean in the caption?). likewise for other figures (I guess it could be repeated or just in Figure 1)

Reviewer #2 (Remarks to the Author):

The manuscript titled 'Boreal conifers maintain carbon uptake with warming despite failure to track optimal temperatures' set out to measure photosynthetic thermal acclimation in mature, field-grown boreal trees, as well as determine whether elevated atmospheric CO₂ interacts with this thermal acclimation. To investigate these questions, the researchers employed an impressive experimental design using open-top whole-tree chambers that allowed for the alteration of CO₂ concentration and air temperature in the air around existing trees. The results reported showed a clear, yet somewhat weaker than expected, thermal acclimation of photosynthesis that seemed to be driven by biochemical adjustments. Interestingly, a doubling of atmospheric CO₂ concentration had limited effects on thermal acclimation of photosynthesis, though there was a key species difference in this respect. I really enjoyed reading this paper, as it was extremely well-written, the experiment well-designed, and the results presented in a way that conveyed a compelling narrative. The data presented not only provides much needed new information regarding thermal acclimation of A in mature trees outside of growth cabinets/glasshouses, but also raises many further interesting questions that these authors and others will hopefully seek to address. I only had two very minor points for the authors to consider prior to publication:

(1) Upon reaching the discussion, I found one of the biggest points that I was hoping to see discussed was what might be driving the observed differences between spruce and tamarack? While there was limited discussion of this (e.g. the authors suggest it could be due to difference in leaf habit), if word count permits I would be intrigued to hear any further thoughts from the authors on the matter. For instance, what would the reasoning be behind a deciduous tree having a weaker suppression of photorespiration? Are there any other ecophysiological differences between these species that could be playing a role?

(2) Given that this paper is focused on acclimation to warming and leaf-level physiology, it would have been nice if leaf temperatures corresponding to the different temperature treatments had been measured and reported in addition to prevailing air temperatures, given the fact that $T_{leaf} \neq T_{air}$. While I don't think this is a crucial oversight for this particular paper, there are a couple of stronger statements made about air temperature exceeding T_{opt} and the implications of this (e.g. Line 205-207), and this is where I think it would be worth at least acknowledging that it is possible the trees that were measured did in fact maintain their T_{leaf} closer to T_{opt} . (I will note that because conifer needles have a far higher surface area:volume ratio than many broadleaved trees, I imagine they would also have a harder time regulating their leaf temperature, and thus perhaps the assumption that $T_{leaf} = T_{air}$ may be more appropriate when discussing conifers.)

The two points above are minor general suggestions that, given journal constraints around word counts, I could understand if the authors are unable to add much more additional content on. I have also included some specific (and very minor) line-by-line suggestions below that are largely just calling a few typos to the authors' attention. Overall, I found this paper to be of high quality, and am pleased to recommend it for publication.

Line 87-88: slightly awkward phrasing in brackets, maybe rearrange to "do some species acclimate while others do not?"

Line 119: I think a word is missing after "3.6 – 4°C" – 'higher' or 'greater', perhaps?

Line 117-122: I would suggest breaking up this sentence into at least two sentences, should help make it easier to read/understand.

Line 124: T_{opt}A is already defined in the introduction so probably don't need to define here again.

Line 179-180: The authors note that responses of A_{opt} and A_gmean were very similar – perhaps worth noting that this suggests that plants largely were able to maintain their T_{opt} close to the prevailing growth temperature (perhaps a further sign of effective thermal acclimation?).

Line 225-227: It could be worth qualifying this statement by noting that this may only apply to plants that are not facing drought stress, as one would think that if VPD and/or soil water availability become challenging the role of stomatal conductance could be just as or more influential on A than biochemical adjustments to temperature. While I appreciate that this manuscript is specifically about the effects of warming, given that warming and drought commonly co-occur it seems appropriate to at least mention.

Line 248: I'm not sure 'Nevertheless' is the right word to start this sentence. Maybe 'By contrast'?

Line 251: change "...is not clear..." to "...are unclear..."

Line 258: Passive voice, consider switching order of sentence (e.g. "Even though prevailing air temperature largely exceeded T_{opt}A")

Line 259: Is "both" a typo here?

Line 281: I would swap that hyphen for a comma

Fig 3: No mention in the results text of the differences in responses in June vs August. If this wasn't mentioned because the difference was minimal than you probably could pool together for this figure. However, if the authors think that the difference associated with seasonal change was noteworthy than I think it would be good to explicitly mention in the results text.

Supplementary Table 3a: Not sure if I'm reading incorrectly, but this table doesn't seem to mention what variable T_{opt} is being tested against (from the text I thought this would be respiration?)

REVIEWER COMMENTS

Reviewer #1 (Remarks to the Author):

Review: Boreal conifers maintain carbon uptake with warming despite failure to track optimal temperatures

General:

This is a really important study from the SPRUCE experiment. The results are also interesting! For the most part, the writing flowed well and interpretation was easy, but I did get bogged down a few times in the details of the results section. The discussion helped with this. I suggest a few things below to help out with the abstract and to help the reader find the most important points. Finally, while I really liked the clear explanation of TBM implications, I think dismissing the need to develop temperature acclimation in models could use a caveat – this was only for two species. There are a lot of other conifers (globally and in North America) that may respond differently. I know we can't experiment on all of them, but perhaps it should be mentioned that this type of trait plasticity is still unknown for non-boreal species (for field grown mature trees).

We thank Reviewer #1 for their positive comments and constructive feedback. The clarifications and additions made to address the comments have helped to improve the manuscript further, and the reviewer's specific comments are addressed below.

As the work was indeed only done on two species, we have revised the text regarding extrapolating the results accordingly (Lines 246 - 258):

“ These new findings with field-grown mature boreal trees agree with an earlier, short-term study with seedlings of the same species²⁴, suggesting that regardless of the experimental approach, life stage, and leaf habit, elevated CO₂ does not have strong effects on the thermal sensitivity of photosynthetic biochemical processes, such as V_{cmax} and J_{max} , in boreal conifers. Since V_{cmax} and J_{max} are key parameters for representing carbon uptake within TBMs⁶, our findings imply that potential interactive effects of elevated CO₂ on temperature sensitivity parameters of V_{cmax} and J_{max} (i.e., their activation energies and thermal optima) can be ignored in TBMs. Our findings also suggest that temperature response functions of these parameters, developed mainly from ambient CO₂-grown plants^{8,40} and currently employed in all TBMs^{6,7,42} might accurately represent carbon uptake for trees growing in both current and projected elevated CO₂ conditions in future climates. However, further research on tree species from other biomes and plant functional types (e.g., broadleaved tree and shrub species) are still needed to validate this conclusion for broad use.”

Abstract:

1. “but these increases were not sufficient to keep pace with warming” – What does this mean? T_{opt} lagged or just changed less? What are the consequences of this? I think both a quantitative example of what this means (0.X per X degrees C) and what the consequences are of T_{opt} not keeping pace would be more powerful.

Thank you for the suggestion – we have now added this information (Lines 50 - 51):

“However, T_{opt} did not keep pace with warming, and only increased 0.26-0.35 °C per 1 °C of warming.”

Regarding the consequences of this observed weak thermal acclimation of T_{opt} , we explain this in the last sentence of the abstract (Lines 53 - 56):

“Although shifts in T_{opt} of these two species are insufficient to keep pace with warming, these boreal conifers can thermally acclimate photosynthesis to maintain carbon uptake in future air temperatures.”

2. Can you indicate early on that Tamarack is deciduous (and maybe say Larch as well)? I don't think everyone knows this.

We agree that this would provide additional useful information, but with a limited word count in the abstract, we could not add this there. However, we have now added this information in the last paragraph of the introduction (Line 133 - 138):

“Here, we assess the thermal acclimation of photosynthesis and its underlying biochemical processes (i.e., V_{cmax} and J_{max}) in mature trees (~ 45 years) of tamarack (also known as larch), a deciduous conifer, and black spruce, an evergreen conifer, exposed to either ambient (hereafter AC) or elevated CO_2 ($\approx + 460$ ppm above ambient; hereafter EC) combined with a warming of up to +9 °C above ambient temperatures in a regression-based design with five temperature treatments (ambient +0, +2.25, +4.5, +6.75, and +9).”

2. I was pretty confused about the ‘modeling’ statement. Part of this was just because I hadn't worked out what you were trying to point out with mean growth temperatures, but the other reason is because I thought you meant you process-based modeled it (like with a TBM or an ecosystem model). If you could point out that the result is from the equations (yes, they are also process-based, but are not a TBM) that would help the reader understand that there is not a major modeling component to the study (like a TBM exercise).

We understand how using the word “model” may imply the use of larger scale models. We have now changed the word “modelled” to “estimated” to reduce potential confusion, but maintain the initial intended meaning (Lines 51 - 53):

“But net photosynthetic rates estimated at the mean growth temperature increased with warming in EC spruce, while remaining constant in AC spruce and in both AC and EC tamarack with warming.”

We have also changed this in the results section (Line 190 - 192):

“We further estimated net CO_2 assimilation at growth temperature conditions for two years (2016 and 2017), representing the entire acclimation period to temperature in this study.”

Results:

1. Lines 154 – 157: had me a bit confused. Both sentences are saying the same thing for both

species but as written, one expects them to be different. Can you just say they both did the same thing and reduce the words? (Yes there is a different range of temperatures for the species but it is the same 'most of the time')

Agreed. We have merged the two sentences as follows (Line 163 - 165):

"In AC-grown tamarack and black spruce, mean daytime growth temperature exceeded T_{optA} ($\Delta MeanT_g > 2$ °C) across all warming treatments (+2.25 to +9 °C) (Figs. 3; Supplementary Table 4)."

2. Tamarack is deciduous (larch?)....does that matter? Deciduous leaves less likely to 'acclimate'? Or would new leaves acclimate? This is explained (one sentence) later but I think it is really interesting that you have these results from a evergreen and deciduous conifer and I would have like a little more discussion about why they may have been expected to respond differently (was there a hypothesis?)

Thank you for this comment. We agree that leaf habit may affect how species respond to warming and elevated CO₂. However, with only one species in each group (deciduous and evergreen), we cannot definitively test the effect of leaf habit in this study. That said, our results do provide a starting point to develop further experiments (with multiple species in each leaf habit group) that explicitly test whether evergreen and deciduous tree species differ in their ability to physiologically acclimate to climate change.

As we mentioned in the Discussion (Line 259 - 278), we measured needle cohorts that developed under the experimental treatments for both species, therefore, the effect of leaf habit on physiological acclimation may have been reduced compared to a study where the evergreen species had pre-existing leaves and the deciduous species did not. But in response to this comment from Reviewer #2, we have further investigated other physiological traits that may have affected the responses to elevated CO₂ between these two species, including stomatal and mesophyll conductance. In this revised version, we have expanded our discussion on the potential role of stomatal and mesophyll conductance for explaining the observed differences between species to elevated CO₂.

"We show that the b parameter was generally increased by elevated CO₂ for both species, suggesting that elevated CO₂ increases the thermal sensitivity of net photosynthesis, a result in line with a shift to photosynthesis being more RuBP-regeneration limited at high CO₂ concentrations (Sage and Kubien, 2008). In addition, elevated CO₂ did affect the T_{optA} , but these effects were species dependent. In tamarack, the T_{optA} was higher in elevated CO₂, which likely reflects a direct suppression of photorespiration^{5,9,37,48}. In contrast, there was no effect of elevated CO₂ on the T_{optA} in black spruce, and these results contrast prior findings in black spruce seedlings²⁴. The reasons behind this lack of elevated CO₂ effect on T_{optA} in mature black spruce are unclear since, similar to tamarack, the needle cohorts that were measured developed in prevailing environmental conditions across treatments. However, the magnitude of suppression of photorespiration by elevated CO₂ may vary across species or plant functional types – or in our case differences in leaf habit (evergreen versus deciduous). In our study, we

cannot make a solid conclusion on the main cause for this, but two possibilities are differences in stomatal and mesophyll conductance between the species. Our data shows little differences in intracellular CO₂ concentration between the two species across the two CO₂ treatments (data not shown), indicating that stomatal limitations are unlikely to underlie the difference in how T_{opt} responds to elevated CO₂. This leaves mesophyll conductance as a possible factor, as higher mesophyll conductance in tamarack could enhance CO₂ supply to Rubisco for a given unit of intercellular CO₂. However, without mesophyll conductance measurements we cannot directly prove this, and future research is needed to investigate this possibility.”

Discussion

3. LINE 205 to 206 is an important point! Maybe put it in the abstract?

We agree that it would have been ideal to include this additional implication of weak thermal acclimation of T_{opt} in the abstract, but with the word limit, it is difficult to incorporate it. As we already mention in the abstract (Line 53 - 56), we did highlight the most direct implication of T_{opt} exceedance by air temperature on carbon uptake at prevailing mean air temperatures across treatment.

4. Lines 239 – 245: Really important point! But, this was only for two species.

We have revised this section (Lines 246 - 258) as noted above.

Figures.

Figure 1: Can there be a legend (instead of just what the symbols mean in the caption?). likewise for other figures (I guess it could be repeated or just in Figure 1)

Thank you for the comment! We tried adding the legend on the figure, but it makes the figure too crowded to interpret easily, so we have kept this information in the figure captions.

Reviewer #2 (Remarks to the Author):

The manuscript titled ‘Boreal conifers maintain carbon uptake with warming despite failure to track optimal temperatures’ set out to measure photosynthetic thermal acclimation in mature, field-grown boreal trees, as well as determine whether elevated atmospheric CO₂ interacts with this thermal acclimation. To investigate these questions, the researchers employed an impressive experimental design using open-top whole-tree chambers that allowed for the alteration of CO₂ concentration and air temperature in the air around existing trees. The results reported showed a clear, yet somewhat weaker than expected, thermal acclimation of photosynthesis that seemed to be driven by biochemical adjustments. Interestingly, a doubling of atmospheric CO₂ concentration had limited effects on thermal acclimation of photosynthesis, though there was a key species difference in this respect. I really enjoyed reading this paper, as it was extremely well-written, the experiment well-designed, and the

results presented in a way that conveyed a compelling narrative. The data presented not only provides much needed new information regarding thermal acclimation of A in mature trees outside of growth cabinets/glasshouses, but also raises many further interesting questions that these authors and others will hopefully seek to address. I only had two very minor points for the authors to consider prior to publication:

We thank Reviewer #2 for their kind words and are grateful for their constructive feedback on our manuscript.

(1) Upon reaching the discussion, I found one of the biggest points that I was hoping to see discussed was what might be driving the observed differences between spruce and tamarack? While there was limited discussion of this (e.g. the authors suggest it could be due to difference in leaf habit), if word count permits I would be intrigued to hear any further thoughts from the authors on the matter. For instance, what would the reasoning be behind a deciduous tree having a weaker suppression of photorespiration? Are there any other ecophysiological differences between these species that could be playing a role?

Thank you very much for this comment. Our previous work in seedlings pointed at different responses in stomatal conductance to warming between these two species. Therefore, we explored the possibility of stomatal conductance differentially affecting the intercellular CO₂ concentration (C_i). However, we did not find any effect of stomatal conductance on C_i between the two species across the two CO₂ treatments. Therefore, this leaves mesophyll conductance as a possible factor in the response of these two species to CO₂. We have now added this information (Line 270 - 278) and updated the entire paragraph (Line 259 - 278) as noted above.

(2) Given that this paper is focused on acclimation to warming and leaf-level physiology, it would have been nice if leaf temperatures corresponding to the different temperature treatments had been measured and reported in addition to prevailing air temperatures, given the fact that T_{leaf} does not necessarily = T_{air}. While I don't think this is a crucial oversight for this particular paper, there are a couple of stronger statements made about air temperature exceeding T_{opt} and the implications of this (e.g. Line 205-207), and this is where I think it would be worth at least acknowledging that it is possible the trees that were measured did in fact maintain their T_{leaf} closer to T_{opt}. (I will note that because conifer needles have a far higher surface area:volume ratio than many broadleaved trees, I imagine they would also have a harder time regulating their leaf temperature, and thus perhaps the assumption that T_{leaf} = T_{air} may be more appropriate when discussing conifers.)

This is a very insightful point, and we agree that ideally we would have measured actual leaf temperatures. However, as noted by the Reviewer, the temperature of small, thin needles (that also have low stomatal conductance!) are well coupled with the surrounding air temperature. Therefore, assuming that leaf temperature is very close to the surrounding air temperature is a reasonable assumption. We have added a sentence in the Results section (Line 161 - 163) to explicitly clarify this point:

“This approach assumes that leaf and air temperatures are similar, a reasonable assumption considering the tight coupling between leaf and air temperature in small leaves⁴³, such as conifer needles.”

The two points above are minor general suggestions that, given journal constraints around word counts, I could understand if the authors are unable to add much more additional content on. I have also included some specific (and very minor) line-by-line suggestions below that are largely just calling a few typos to the authors’ attention. Overall, I found this paper to be of high quality, and am pleased to recommend it for publication.

Thank you very much again for highly recommending our paper for publication in Nature Communications. Your suggestions substantially improved our manuscript.

Line 87-88: slightly awkward phrasing in brackets, maybe rearrange to “do some species acclimate while others do not?”

Revised accordingly!

Line 119: I think a word is missing after “3.6 – 4°C” – ‘higher’ or ‘greater’, perhaps?

Thanks for noticing this. We have added the word “higher”

Line 117-122: I would suggest breaking up this sentence into at least two sentences, should help make it easier to read/understand.

We have broken this statement into two sentences (Lines 121 - 125):

“In the same study, warming increased T_{optA} by 0.36 – 0.65 °C per 1 °C warming regardless of CO₂ treatments. But elevated CO₂-grown seedlings had a T_{optA} that was generally 3.6 – 4 °C higher than their ambient CO₂-grown counterparts when measured at prevailing growth CO₂, likely due to direct suppression of photorespiration by elevated CO₂.”

Line 124: T_{optA} is already defined in the introduction so probably don’t need to define here again.

We have changed the text accordingly.

Line 179-180: The authors note that responses of A_{opt} and A_{gtmean} were very similar – perhaps worth noting that this suggests that plants largely were able to maintain their T_{opt} close to the prevailing growth temperature (perhaps a further sign of effective thermal acclimation?).

Thanks for the suggestion. We have added a sentence to emphasize this (Line 189 - 190), as suggested:

“These results suggest that, overall, the two species were able to maintain carbon uptake at prevailing growth temperatures.”

Line 225-227: It could be worth qualifying this statement by noting that this may only apply to plants that are not facing drought stress, as one would think that if VPD and/or soil water availability become challenging the role of stomatal conductance could be just as or more influential on A than biochemical adjustments to temperature. While I appreciate that this manuscript is specifically about the effects of warming, given that warming and drought commonly co-occur it seems appropriate to at least mention.

Agreed. We have revised the sentence accordingly (Line 235 - 239).

“These results imply that changes in photosynthetic biochemical processes strongly underlie the adjustment of photosynthesis to long-term changes in growth temperature, regardless of experimental approach or tree life stage, although stomatal limitations are likely to play a greater role in limiting photosynthesis in water-stressed trees.”

Line 248: I’m not sure ‘Nevertheless’ is the right word to start this sentence. Maybe ‘By contrast’?

We have changed the wording accordingly.

Line 251: change “...is not clear...” to “...are unclear...”

Done.

Line 258: Passive voice, consider switching order of sentence (e.g. “Even though prevailing air temperature largely exceeded T_{optA} ”)

We have revised accordingly.

Line 259: Is “both” a typo here?

Thanks for noticing this. We have deleted the word “both”

Line 281: I would swap that hyphen for a comma

Done.

Fig 3: No mention in the results text of the differences in responses in June vs August. If this wasn’t mentioned because the difference was minimal than you probably could pool together for this figure. However, if the authors think that the difference associated with seasonal change was noteworthy than I think it would be good to explicitly mention in the results text.

Thanks for the comment! We have revised the figure as suggested by merging both June and August since they were not statistically significant.

Supplementary Table 3a: Not sure if I'm reading incorrectly, but this table doesn't seem to mention what variable T_{opt} is being tested against (from the text I thought this would be respiration?)

We apologize for the confusion, we agree that the tables (Supplementary Tables 3a and b) were not initially clear. We have now revised these two tables to specifically indicate the photosynthetic parameters are being tested for their sensitivity to temperature and CO_2 treatments.